# The Mixing Ratio and Filling-Amount Affect the Tissue Browning and Antioxidant Properties of Fresh-Cut Baby Leaf Lettuce (*Lactuca sativa* L.) and Rocket (*Eruca sativa* Mill.) Grown in Floating Growing Systems

**DOI:** 10.3390/foods11213515

**Published:** 2022-11-04

**Authors:** Lijuan Zhan, Roberta Bulgari, Giuseppe Pignata, Manuela Casale, Silvana Nicola

**Affiliations:** 1College of Food Science and Technology, Henan Agricultural University (HAU), Wenhua Road 95, Zhengzhou 450002, China; 2Department of Agricultural, Forest and Food Sciences (DISAFA), Vegetable Crops & Medicinal and Aromatic Plants, VEGMAP, University of Turin, UNITO, Largo Paolo Braccini 2, 10095 Grugliasco, TO, Italy

**Keywords:** leafy vegetables, quality, phytochemicals, microbial contamination, postharvest decay, shelf-life

## Abstract

Different types of baby leaf vegetables (BLV) are often mixed and packaged as salad mixes. This work has evaluated the effects of BLV mixing ratios (100% lettuce ‘Lollo Bionda’, 100 LB; 75% lettuce + 25% rocket, 75 LB; 50% lettuce + 50% rocket, 50 LB) and the weight filling amount (125 g filling amount, 125F; 250 g, 250F) on the antioxidant properties and browning potential (BP) of lettuce and rocket baby leaves during storage for 9 days at 4 °C in the dark. The samples were packaged in thermos-sealed bags previously prepared using polypropylene film. The results showed that the 50 LB mix had preserved high amounts of chlorophylls and internal nutrients on d9, regardless of the filling amount. No visible browning symptoms were detected in the 50 LB samples. The 50 LB × 125F mix was found to be the most efficient strategy to maintain the antioxidant property of BLV. Thus, the optimisation of the mixing ratio and its combination with an appropriate filling amount could represent an effective postharvest practice.

## 1. Introduction

Baby leaf vegetables (BLV) is the term used to define leafy vegetables that are harvested at a very early stage of growth and processed when they have young leaves [1,2], such as green leaf lettuce (*Lactuca sativa* L. var. *crispa* cv. Lollo Bionda, hereafter: LB) and cultivated rocket (*Eruca sativa* Mill., hereafter: RO). The production of this type of product is increasingly being carried out by means of soilless cultivation systems (SCSs), which represent an efficient way of managing cultivation inputs and resources and allow a high product quality to be obtained [3,4]. BLV are rapidly becoming an important segment of the fresh-cut industry, thanks to their convenience, high value, unique sensory characteristics, and healthy phytochemicals, which are sources of minerals, vitamins, and antioxidants [1,5,6,7]. Therefore, the development of BLV is recommended because of their possible health benefits and excellent processing characteristics [6]. Nevertheless, BLV are highly perishable and susceptible to quality decay along the supply chain as their tender immature leaves have a higher respiration rate and active metabolism than fully mature ones [8,9]. The respiration and active metabolisms of BLV can subsequently trigger a significant deterioration of the colour, texture, and nutritional qualities of these young leaves [10]. Thus, innovative and effective strategies are urgently needed to preserve their quality traits and prolong the shelf-life.

Besides lowering the storage temperature, modified atmosphere packaging (MAP) is considered to be the most effective practice adopted to prolong the shelf-life of fresh and minimally processed produce [10]. MAP protects fresh produce from damage during distribution and selling and provides consumers with a convenient and efficient unit, but more importantly, it creates a favourable modified atmosphere which preserves fresh produce [11,12]. Passive modified atmosphere packaging (passive MAP), whereby packages are sealed in air, is considered to be a suitable packaging method for BLV. The composition and changes over time of the gases in passive MAP are mainly related to the gas permeability of the package film, to respiration of the product, and to the product filling amount [13]. Hence, the product filling amount in a given-size package impacts any change in the gas dynamics within the package, and thus in turn influences the postharvest quality of the product. Different filling amounts lead to different scenarios. Generally, increasing the filling amount reduces the free headspace volume and modifies the equilibrium of the atmosphere rapidly [14]. The greater the filling amount in a given package is, the more intensive and faster equilibrium of the atmosphere occurs in the headspace; this is because a larger mass of vegetables consumes more O_2_ and produces more CO_2_. At the same time, the increase in the filling amount reduces the free volume of the headspace, and thus increases the potential physical damage of vegetable tissue because of the higher compression rate. Vice versa, reducing the filling amount reduces the possibility of compression injury because of the larger free volume. However, such a reduction makes the vegetable tissue reach gas equilibrium more slowly, with a higher availability of O_2_ in the headspace, which in turn accelerates respiration and, subsequently, an undesirable phytochemical oxidative degradation [13]. Therefore, the optimum filling amount in a given-size package is critical to preserve the postharvest quality of fresh produce.

Moreover, today’s consumers prefer fresh products with diverse tastes, textures and dietary values. To meet this demand, different types/varieties of BLV are often mixed and packaged to make products more colourful and attractive. It is worth noting that the mixing a ratio of different types/varieties of BLV in the same package has been shown to affect the inherent quality and storage characteristics of products to a great extent [6]. One of our previous studies showed that the mixing ratio of fresh-cut baby lettuce with 50% green lettuce + 50% red lettuce and 25% green lettuce + 75% red lettuce had higher inherent quality on d1 than other mixing ratios [6]. This positive effect might be attributable to the synergistic effect of the different leaf types and their mixing ratios. Different leaf types have different respiration rates, and, when at the optimal mixing ratio, they can rapidly create a desirable gas equilibrium in the headspace. Therefore, adopting an optimal mixing ratio of different types of BLV in the same salad bag could be of interest for consumers, and represents a positive postharvest practice to preserve the inherent and commercial quality characteristics of BLV throughout the shelf-life [6].

Baby lettuce (*Lactuca sativa* L.) and cultivated rocket (*Eruca sativa* Mill.) are typical BLVs and have recently been used increasingly in mixed salads [6,15,16,17]. Lettuce is among the most consumed leafy vegetables in the world [18,19,20,21]. In particular, baby lettuce leaves represent an innovative vegetable produce to meet modern consumers demand [17]. Most salads, at present, continue to be mainly composed of lettuce.

Rocket is particularly popular and marketed either on its own or mainly in mixed salads, in combination with other leafy vegetables [22] because of its spicy taste and sesame-scented leaves [23,24].

Previous studies we conducted revealed that the filling amount in a given-size bag significantly affected the pigment content of minimally processed rocket [15,25]. We have recently demonstrated that different mixing ratios of green and red baby-leaf lettuces in the same salad bag markedly influenced the inherent and commercial quality characteristics [6]. Lettuce and rocket are often commercially mixed at different mixing ratios and with different filling amounts. These differences possibly affect their postharvest storage quality and shelf-life, as they have different postharvest properties and biological behaviour, for example, different respiration rates. Earlier research studies were mainly focused on the postharvest properties of either lettuce or rocket on their own [25]. However, to date, no information is available concerning the effect of the mixing ratio and packaging filling amount on the postharvest characteristics of mixed lettuce and rocket salads throughout their shelf-life.

The purpose of the present study was to investigate the influence of different weight mixing ratios (100 LB, 100% lettuce; 75 LB, 75% lettuce + 25% rocket; 50 LB, 50% lettuce + 50% rocket) and filling-amounts (125F, 125 g; 250F, 250 g) on the antioxidant properties and tissue browning of mixed lettuce and rocket baby leaves during 9 days of storage at 4 °C in the dark. Parameters related to the antioxidant properties (total phenolic-TP, ascorbic acid-AA, dehydroascorbic acid-DHAA, antioxidant capacity-AC) and tissue browning (browning potential-BP, soluble quinone-SQ, polyphenol oxidase-PPO, phenylalanine ammonia lyase-PAL) were analysed on days 1 (d1) and 9 (d9) of storage. In addition, the headspace gas composition (O_2_%, CO_2_%), pigment content, fresh weight loss, ion and salt contents (NO_3_^−^, PO_4_^3−^, and CaCO_3_), as well as the total bacterial (TB), yeast + mould (Y + M) counts were assayed during storage. At harvest (d0), some determinations were carried out with regard to the raw material of the two species separately, lettuce and rocket, to assess the phytochemical composition of the different species.

## 2. Materials and Methods

### 2.1. Plant Material and Growing Conditions

The raw material was obtained from cultivations carried out in the Tetti Frati Experimental Centre of DISAFA (44°53′11.67″ N; 7°41′7.00″ E-231 m a.s.l. Carmagnola (TO), Italy) from September to October in an automatically controlled temperature greenhouse. Green leaf lettuce (*Lactuca sativa* L. var. *crispa* cv. Lollo Bionda, hereafter: LB) (Furia Seed S.r.l., Monticelli Terme (PR), Italy) and cultivated rocket (*Eruca sativa* Mill., hereafter: RO) (Olter S.r.l., Asti, Italy) were sown at a density of ca. 784 plants/m^2^ and ca. 1961 plants/m^2^, respectively in 60-cell multicellular trays filled with a specific commercial peat-based horticultural medium (NeuhausHuminsubstrat N17; Klasmann-Deilmann^®^ GmbH, Geeste, Niedersachsen, Germany), prepared on the Azienda Agricola Vivaistica Ricca Sebastiano farm (Carignano (TO), Italy). Plants were grown in a lab-scale pilot plant (LSPP) for Floating Growing Systems (FGS), as described by Nicola et al. [26], using a 40/60 N-NO_3_^−^/N-NH_4_^+^ HNS composed of (all in mmol L^−1^): 6 N, 2 P, 6 K, 2 Mg and 2.5 Ca. The LB was grown in a Floating System (FL), while the RO was grown in an Ebb-and-Flow System (EF), due to its high sensitivity to hypoxia. The EF system was scheduled to have 3 days of flotation and 2 days of ebb. Harvesting took place early in the morning to avoid the hottest hours of the day, 41 days after planting for LB and 32 days for RO, using sanitised tools. After harvest, the raw material was immediately transferred to the postharvest laboratory, which is at the same site as the LSPP, to be analysed and processed as a fresh-cut product.

The statistical experimental design was a randomised complete block design (RCBD). A single factorial experiment design (2 species × 3 blocks) was adopted during the growing period. Ten whole seeded trays were considered to represent the experimental unit for each block. The postharvest experiment consisted of 3 mixing ratios × 2 filling amounts × 3 blocks.

### 2.2. Processing, Packaging, and Storage Conditions

The raw material was sorted in a cold temperature room, and any damaged or yellow leaves were discarded. Three mixing ratios of LB and RO were considered: 100 LB (100% lettuce), 75 LB (75% lettuce + 25% rocket), and 50 LB (50% lettuce + 50% rocket). These mixed samples were subsequently packaged in two filling amounts of F125 (125 g) and F250 (250 g), using 0.25 m × 0.35 m thermos-sealed bags previously prepared using polypropylene film (20 µm thick, 18.2 g m^−2^ weight, with a permeance to oxygen of 1990 cm^3^ m^−2^ d^−1^ bar^−1^, to carbon dioxide of 7800 cm^3^ m^−2^ d^−1^ bar^−1^, and to water vapour of 5.8 g m^−2^ d^−1^ bar^−1^, Alvapack S.r.l., Bologna, Italy). A total of 36 samples were packed. The packaged samples were stored for 9 days in refrigerated chambers (MEDIKA 600; C.F. di Ciro Fiocchetti & C. S.n.c., Luzzara, (RE), Italy) at 4 °C, in the dark. All the qualitative parameters were measured after 1 day of storage (d1) and at the end of storage (d9), except for the total TB and Y + M, which were only analysed on d9.

### 2.3. Headspace Gas and Fresh Weight Loss Assay

The headspace gas composition (O_2_% and CO_2_%) in the bags was measured using a Check point Handheld Gas Analyser (PBI-Dansensor AS, Ringsted, Denmark). The fresh weight loss (FWL) was measured daily by weighing the bags during storage, and it was calculated progressively as a percentage decay index of freshness from harvest (d0) onwards.

### 2.4. Pigment Content Analysis

The chlorophyll *a* (Chl. *a*), chlorophyll *b* (Chl. *b*) and carotenoid (*Car*.) contents were determined through destructive chemical extraction, according to the protocol suggested in literatures [25,27], using 0.25 g of fresh leaves. The results were expressed as mg g^−1^ fresh weight (FW).

### 2.5. Antioxidant Property Parameter Assay

The total phenolic (TP) content was determined using the Folin–Ciocalteu procedure [28], using 1 g of sample. The ascorbic acid (AA) and dehydroascorbic acid (DHAA) contents were determined using 10 g of leaves, according to the method developed by Kampfenkel et al. [29]. The results were expressed as mg g^−1^ FW on the basis of the standard curves produced by freshly prepared L-ascorbic acid. The antioxidant capacity (AC) was determined using the ferric reducing ability of plasma (FRAP) assay, according to the Benzie and Strain method [30]. The AC values were calculated from a standard curve produced by fresh ammonium ferrous sulphate hexahydrate and were expressed as µmol Fe^2+^ g^−1^ FW.

### 2.6. Tissue Browning Parameter Assay

The browning potential (BP) and soluble o-quinone (So-Q) content were determined according to Couture et al. [31] and Tardelli et al. [32]. The polyphenol oxidase (PPO) and phenylalanine ammonia lyase (PAL) activities were assessed according to Degl’Innocenti et al. [33] and Zhan et al. [25].

### 2.7. Tissue Ion and Salt Content Assay

The nitrate, phosphate, and calcium carbonate (NO_3_^−^, PO_4_^3−^, and CaCO_3_) contents were determined as previously described [6], analysing 10 g of each sample. The assay was carried out using a refractometric kit (Merck Reflectoquant RQflex2^©^; Merck KGaA, Darmstadt, Hessen, Germany), according to the manufacturer’s instructions. The data were expressed as mg g^−1^ FW.

All the spectrophotometric analyses were conducted using a Beckman DU^®^-65 spectrophotometer (Beckman Coulter Inc., Fullerton, CA, USA).

### 2.8. Microbial Analysis

The total bacterial (TB) and yeast + mould (Y + M) counts were determined on 25 g of fresh tissue per sample in Petri dishes, using a selective substrate after incubation at 30 °C for 48 h [34,35,36].

### 2.9. Statistical Analysis

The harvest and postharvest data were submitted to analysis of variance (ANOVA) using the Statistical Package for Social Science (SPSS Inc., Chicago, IL, USA). When ANOVA was significant, the species, mixing ratios, and filling amount effects were tested using the F-test, and the mixing ratio × filling amount effect was tested using Tukey’s multiple range test.

## 3. Results

### 3.1. The Headspace Gas and Fresh Weight Loss

The O_2_ and CO_2_ levels were significantly influenced by the mixing ratio, filling amount, and the interaction between the mixing ratio × filling amount on d1, but not on d9 (Table 1). Specifically, 100 LB × F125 had the highest (19.43%) O_2_ and the lowest (1.1%) CO_2_ levels in the bag headspace on d1; on the other hand, 75 LB × F250 showed the lowest (15.23%) O_2_ and the highest (4.83%) CO_2_ levels on d1 (Table 1). The average values of O_2_ and CO_2_ in the headspace were 15.29% and 4.13% on d9, respectively. The FWL in all the samples increased over time, as expected, and was only influenced significantly by the filling amount on d1 of storage (*p* = 0.032) (data not shown). The FWL was higher in F125 (0.04%) than in F250 on d1 (0.01%). The maximum FWL was 0.75%, and it occurred in the 50 LB × F125 samples on d9, although all the leaves were still fresh and acceptable for the market.

### 3.2. Pigment Content

The pigment content in RO was markedly higher at harvest (d0) than that of LB (Table 2). During postharvest, the pigment content was influenced significantly by the mixing ratio, the filling amount, and the interaction of mixing ratio × filling amount on d1, except for the Carotenoid content (Car.), which was not significantly influenced by the filling amount (Table 1). The 50 LB × F250 had the highest pigment content on d1. The pigments were only significantly influenced by the mixing ratio on d9. The highest pigment concentration on d9 was found in 75 LB and 50 LB and was statistically different from 100 LB (Table 1).

### 3.3. Antioxidant Property Parameters

#### 3.3.1. Antioxidant Capacity

At harvest (d0), the species had no significant effect on AC (data not shown), whose average value was 3.88 μmol Fe^2+^ g^−1^ FW. AC was only influenced significantly by the mixing ratio treatment on d1 and the interaction between the mixing ratio × filling amount on d9, respectively (Table 3). The highest AC was found in 50 LB on d1, and it was statistically higher than 75 LB and 100 LB. The 50 LB × F125 showed the highest AC value on d9, while 75 LB × F125 showed the lowest (Table 3).

#### 3.3.2. Total Phenolic Content

At harvest (d0), the species had a significant effect on the TP content, with a higher TP contents in RO than in LB (Table 2). During postharvest, the TP content was only influenced significantly by the mixing ratio on d1, while it was influenced by the mixing ratio, filling amount, and their interaction on d9 (Table 3). The highest TP content on d1 was found for 50 LB, which was statistically different from all the other leafy mixes. The TP content in the 50 LB × F125 samples was the highest on d9, while that in 100 LB × 250F was the lowest (Table 3).

#### 3.3.3. Ascorbic Acid and Dehydroascorbic Acid Contents

At harvest (d0), the species had a significant effect on the AA content, which was 1.65-fold higher in RO than in LB (Table 2). The species had no significant effect on the DHAA content (data not shown). The average value of DHAA was 0.02 mg g^−1^ FW. During postharvest, the AA content was influenced significantly by both of the main factors on d1 and d9, while the DHAA content was only influenced significantly by the mixing ratio on d1 and d9 (Table 3). The AA content in 50 LB was significantly higher than that of 75 LB, while that of 75 LB was significantly higher than that of 100 LB on d1 and d9 (Table 3). The AA content in 250F was higher than that of 125F on d1, while it showed an opposite trend on d9. The highest DHAA content was observed in 50 LB on d1, and it was significantly higher than that of 100 LB, but not than that of 75 LB. The highest DHAA content on d9 was found in 50 LB, and it was significantly higher than all the other leafy mixes (Table 3).

### 3.4. Tissue Browning Parameters

#### 3.4.1. Browning Potential and So-Q Content

At harvest (d0), the species had no significant effect on either the BP or the So-Q contents (data not shown), whose average values were 0.66 Abs_340_ and 0.11 Abs_437_, respectively. After harvest, both the BP and the So-Q contents were significantly influenced by the mixing ratio on d1 and d9 (Table 4). In addition, BP was also influenced significantly by the filling amount on d9 (Table 4). The highest BP and So-Q contents were found in 50 LB and the lowest in 100 LB, on both d1 and d9. The 75 LB was statistically different from 50 LB and 100 LB, except for the So-Q content on d9. BP was higher in F125 than in F250 on d9 (Table 4).

#### 3.4.2. Enzyme Activity

At harvest (d0), the PAL activity in RO was 3.75-fold higher that in LB (Table 2). The PPO activity was not statistically different in these two species (data not shown). The average value of PPO activity was 13.48 PPO Units g^−1^ FW. The PAL and PPO activities were both influenced significantly by the mixing ratio on d1, and by the main factors and their interaction on d9 (Table 4). The PAL activity in 50 LB was significantly higher than in 75 LB and 100 LB on d1, and there was no significant difference between the latter two. The PAL activities on d9 were higher in the 50 LB × F125 samples and lower in both the 100 LB × F125 and 100 LB × F250 samples, and the latter two had the same PAL values. The highest PPO activity was found in 50 LB on d1, and this result was statistically different from 75 LB, but not from 100 LB. The PPO activities were higher in the 100 LB × F250 samples and lower in the 100 LB × F125 samples on d9 (Table 4).

### 3.5. Ion and Salt Contents of the Tissue

At harvest (d0), there was no significant difference in the nitrate concentration between species (data not shown), and the average value in both species was ca. 0.48 mg g^−1^ FW (data not shown). The species had a significant effect on the PO_4_^3−^ and CaCO_3_ contents; the former was higher in LB than in RO, while the latter was lower (Table 2). During postharvest, the NO_3_^−^ content was not influenced significantly by any treatment on d1, while it was influenced by the mixing ratio on d9, with 50 LB having the highest NO_3_^−^ content (Table 5). The PO_4_^3−^ content was influenced significantly by the mixing ratio on d1, and by the mixing ratio and by the mixing ratio × filling amount interaction on d9. The PO_4_^3−^ content in 100 LB and 75 LB did not statistically differ on d1, and were significantly higher than that of 50 LB. Both 100 LB × F250 and 100 LB × F125 had the same higher PO_4_^3−^ content on d9, while 50 LB × F250 had the lowest PO_4_^3−^ content. The CaCO_3_ content was influenced significantly by the mixing ratio on d1 and by both of the main factors on d9 (Table 5). The CaCO_3_ content in 50 LB was significantly higher than that of 75 LB and 100 LB on both d1 and d9. The F125 had a higher CaCO_3_ content than F250 at the end of storage (Table 5).

### 3.6. Microbial Analysis

At harvest (d0), there was no significant difference in the TB count and Y + M count between the two species (data not shown), which showed average values of 7.27·10^3^ colony-forming units (cfu) g^−1^ FW and 1.14·10^2^ cfu g^−1^ FW, respectively. The TB count and Y + M count were only influenced significantly by the mixing ratio at the end of storage (Table 5). The highest TB count was found in 75 LB, which resulted to be statistically different from 100 LB, but not from 50 LB, which in turn was not statistically different from 100 LB. The highest Y + M count was found in 75 LB and 50 LB, both of which were significantly higher than that of 100 LB (Table 5).

## 4. Discussion

SCSs guarantee flexibility and crop intensification [37], and also provide high yields and good quality leafy vegetables, such as lettuce (*Lactuca sativa* L.) [6], garden cress (*Lepidium sativum* L.) [38], corn salad (*Valerianella olitoria* L.) [39], and rocket (*Eruca sativa* Mill.) [39]. In this study, our results have revealed that SCSs have a high fresh leaf production (LB: 2005.00 g m^−2^; RO: 889.60 g m^−2^) of lettuce and rocket, with high levels of phytochemicals, as represented by pigments, TP, AA, and AC. This is in agreement with a previous study that showed SCSs produced higher lettuce (2054 g m^−2^) and rocket (1684 g m^−2^) yields, with higher nutritional quality than traditional culture systems (TCS) [6,40]. In addition, this culture system allows both LB and RO to have a short growing cycle (41 days for LB, 32 days for RO) and produces clean raw material at harvest. In fact, the cultivation through soilless systems allows obtaining a cleaner and hygienically safer product than traditionally soil grown systems [40].

The NO_3_^−^ content in both species was ca. 0.48 mg g^−1^ FW (data not shown) at harvest (d0), which is very low and well below the maximum nitrate content of lettuce and rocket allowed by the EU (Commission Regulation No. 1258/2011). Therefore, the floating growing system in this study could be considered an innovative and efficient cultural technology to produce BLV with high nutritional quality, a short cultural cycle, less soil and microbial pollution, and a low nitrate content for the fresh-cut chain. However, the quality and safety of fresh produce not only depend on good cultural practices and growing systems, but also on appropriate postharvest practices. Yellowing caused by chlorophyll degradation is one of the most important colour deteriorations of BLVs during postharvest storage. This undesirable change could be delayed by adopting 50 LB as shown in the present study. In fact, 50 LB showed a significantly higher chlorophyll content than 100 LB during storage (Table 1). This can probably be attributed to the higher pigment content in RO than in LB at harvest (Table 2). In this study, the higher the RO ratio was, the higher the pigment content of the mixed sample, thus confirming that the success of BLV preservation starts with high-quality raw materials. In addition, it should be noted that during storage, Chl *b* decreased quickly in almost all the samples, while Chl *a* decreased slowly and even increased slightly in some samples (Table 1). In fact, during storage, the average content of Chl *b* decreased by 14.29%, while Chl *a* increased by 4.76% from d1 to d9, regardless of the treatment (Table 1), thus indicating Chl *b* degraded more than Chl *a*. This result is consistent with the chlorophyll degradation mode, whereby Chl *b* first converts into Chl *a*, enters the degradation pathway [41], and then results in an apparent retardation of the decline in the Chl *a* content. A similar chlorophyll degradation pattern was also found for other BLVs, such as spinach [42]. Apart from yellowing, tissue browning is another colour deterioration mechanism that limits the shelf-life and marketability of BLVs. In this study, 50 LB resulted in a higher accumulation of BP and So-Q than 100 LB during storage, thus indicating that the samples were more susceptible to browning when the RO ratio increased. Nevertheless, no visible browning symptoms were detected in the 50 LB samples at the end of storage. This is possibly associated with the high AA content in 50 LB, as a high level of AA might have been involved in inhibiting the occurrence of browning of the rocket salad tissue by reducing quinones to TP compounds [43]. According to the biochemical model of tissue browning, TP compounds are oxidised to quinones by PPO in the presence of oxygen. The increase in So-Q in the 50 LB samples theoretically resulted in the consumption and decline of TP. However, the TP content did not decrease, but remained constant during 9d of storage (Table 3). Thus, it has been hypothesised that the consumed TP was possibly partially compensated through a reduction in quinones, as induced by AA as well as de novo synthesis. A significantly positive correlation (R = 0.549, *p* = 0.001) between AA and TP (Appendix A) indicated that a high level of AA might have been involved in the reduction in TP from quinones. AA is spontaneously oxidised to DHA in the reduction of quinones to phenols. A reaction in the opposite direction that regenerates AA is not the most favourable and would lead to a decrease in AA and an increase in DHA. The AA in the 50 LB samples, which conformed to this mode, decreased by 21.77%, while DHA had increased by 107.5% at the end of storage (Table 3). However, this result was inconsistent with the findings that both the AA and DHA contents in rocket salad decreased during storage [43]. The authors attributed such a DHA decrease to the spontaneous conversion of DHA to diketogulonic acid.

Improvement and retention of the antioxidant properties of postharvest fresh fruit and vegetables has become an important issue in recent years [44], as a large number of epidemiological studies have confirmed that natural antioxidants, including vitamins, carotenoids, and TP compounds, play important roles in human health, especially in combating some diseases [45]. For this reason, both the World Health Organisation (WHO) and the Food and Agricultural Organisation (FAO) recommend a minimum of 400 g of fruit and vegetables per day. In this study, the 50 LB × 125 F treatment had preserved high levels of AC and TP at the end of storage. The AA content in these samples was 57.38% higher than the mean of all the other samples after storage (Table 3). This favourable synergistic effect might be associated with the initial high nutritional quality, low respiration rate, and less crush injury of the samples during storage. As mentioned above, RO had higher levels of antioxidants (TP, AA) than LB at harvest (d0) (Table 2). The high mixing ratio of RO resulted in a high initial antioxidant capacity of the 50 LB × 125 F samples. In addition, the lower respiration rate of the 50 LB × 125 F samples during storage (O_2_, 15.70% vs. 15.29%, CO_2_, 3.47% vs 4.13% on d9 of storage) than the mean of all the other samples favoured the alleviation of the oxidation degradation of TP and AA. Such a low respiration rate of the 50 LB × 125 F samples was possibly caused by the lower filling amount of fresh material and the decrease in crushing mechanical injury, which can trigger a rapid respiration increase in fresh produce. These findings, which were similar to our previous studies, demonstrated that a reduced filling amount (50 g vs. 100 g) could have reduced the chlorophyll degradation of rocket as a result of a lower compression effect inside the package [15].

It is worth noting that the TP content in the 50 LB × 125 F samples increased during storage (Table 3), thus indicating a de novo synthesis of TP. This could be explained by considering the significantly high PAL activity in the 50 LB × 125 F samples at the end of storage (Table 4). TP biosynthesis takes place via the phenylalanine ammonia pathway, in which PAL, a key enzyme responsible for TP synthesis, catalyses the biotransformation of L-phenylalanine to trans-cinnamic acid, and this subsequently leads to the production of TP [33]. TP biodegradation involves the catalytic oxidation of PPO in the presence of oxygen. Therefore, the actual assayed TP content in the samples depends on the balance between its biosynthesis and oxidative degradation [33,42]. Overall, considering the variation trend of the TP, PAL, and PPO in the 50 LB × 125 F samples, we speculated that TP changed dynamically as a result of the catalytic synthesis of PAL and the oxidative degradation of PPO, as both PAL and PPO enzyme activities were detected and increased over time. On the other hand, on d9, the PAL activity was significantly high and 1.73-fold higher than the mean value, while the PPO activity was almost as low as the mean value. Therefore, the increase in TP in the 50 LB × 125 F samples possibly resulted from the difference between the greater de novo synthesis of TP catalysed by the high PAL activity and the lower oxidative degradation due to the low PPO activity. This speculation was also sustained by a significantly positive correlation (R = 0.596, *p* < 0.001) between the TP content and the PAL activity (Appendix A). Contradictory results have often been reported in publications concerning the AC of fresh fruit and vegetables, as the antioxidant properties of natural phytochemicals are multifunctional. The AC of most BLVs is mainly derived from phytochemicals, among which are TP, AA, flavonoids, and carotenoids [42]. In this study, TP and AA seemed to be the main contributors to AC, because both were positively correlated to the AC, regardless of the treatments (Appendix A).

## 5. Conclusions

SCS is an innovative and efficient cultivation technology that can be used to produce lettuce and rocket salad of high nutritional quality, with a short cultural cycle, less soil and microbial pollution, and high added value for the fresh-cut chain. A mixing ratio of 50 LB has been found to favour the preservation of chlorophyll and the internal nutrients during cold storage. No visible browning symptoms were detected in the 50 LB samples, in spite of its high BP and So-Q contents, which might be due to the high AA content in the samples. A combination of 50 LB × 125F was an effective strategy to maintain the antioxidant capacity of BLV as it maintained high AC, TP, and AA contents, with the latter two being the main contributors to AC. This might be due to the high quality of the initial raw material and the reduced compression injury. Overall, this study suggests both efficient cultural practices to obtain high quality BLVs and optimal mixture ratio and packing filling amount for fresh-cut salads. In future studies, other factors, such as fluctuations in the storage temperature and light irradiation, will be investigated and other salad mixes will also be studied in further depth. Thus, an optimum synergistic effect of pre- and postharvest practices/factors will help to guarantee the quality and safety of BLVs from the farm to the table.

## Figures and Tables

**Table 1 foods-11-03515-t001:** Gas composition in the headspace of the bags and pigment content of the fresh-cut baby leaf lettuce and rocket during shelf-life.

	O_2_ (%)		CO_2_ (%)		Chl. *a* (mg g^−1^ FW)		Chl. *b* (mg g^−1^ FW)		Car. (mg g^−1^ FW)	
	d1	d9	d1	d9	d1	d9	d1	d9	d1	d9
Mix										
100 LB	18.75 a ^z^	16.65	1.85 b	3.10	0.18 b	0.11 b	0.06 b	0.04 b	0.08 b	0.04 b
75 LB	16.80 b	12.93	3.43 a	5.83	0.20 ab	0.26 a	0.07 b	0.07 a	0.09 b	0.10 a
50 LB	16.72 b	16.28	3.58 a	3.47	0.27 a	0.29 a	0.09 a	0.08 a	0.12 a	0.11a
Filling amount										
F250	16.53 b	14.73	3.78 a	4.79	0.24 a	0.21	0.08 a	0.06	0.10	0.08
F125	18.31 a	15.84	2.13 b	3.48	0.19 b	0.23	0.06 b	0.07	0.08	0.09
Mix × Filling amount										
100 LB × F250	18.07	15.83	2.60	3.80	0.17	0.11	0.05	0.03	0.08	0.04
100 LB × F125	19.43	17.47	1.10	2.40	0.18	0.11	0.06	0.05	0.08	0.04
75 LB × F250	15.23	11.50	4.83	7.10	0.22	0.22	0.07	0.06	0.09	0.08
75 LB × F125	18.37	14.37	2.03	4.57	0.19	0.30	0.06	0.08	0.09	0.12
50 LB × F250	16.30	16.87	3.90	3.47	0.34	0.31	0.11	0.09	0.14	0.11
50 LB × F125	17.13	15.70	3.27	3.47	0.20	0.27	0.07	0.07	0.09	0.10
Mean	17.42	15.29	2.96	4.13	0.21	0.22	0.07	0.06	0.09	0.08
SE	0.45	2.15	0.37	1.57	0.03	0.03	0.01	0.01	0.01	0.01
Significance										
Mix	*<0.001*	0.181	*<0.001*	0.186	*0.006*	*<0.001*	*0.001*	*<0.001*	*0.005*	*<0.001*
Filling amount	*<0.001*	0.531	*<0.001*	0.315	*0.021*	0.610	*0.025*	0.307	0.061	0.434
Mix × Filling amount	*0.040*	0.634	*0.025*	0.724	*0.033*	0.192	*0.013*	0.160	*0.022*	0.152

d1 = 1 day of storage; d9 = 9 days of storage; LB = percentage of ‘Lollo Bionda’ in the mix with rocket; F = grams of the filling amount in the fresh-cut salad bag; O_2_ = oxygen; CO_2_ = carbon dioxide; Chl. *a* = chlorophyll *a* content; Chl. *b* = chlorophyll *b* content; Car. = carotenoid content; FW = fresh weight; SE = standard error. ^z^ Mean separation (in columns) by Tukey’s multiple range test at *p* ≤ 0.05 for the Mix, and by the F-test at *p* ≤ 0.05 for the Filling amount. The Mix × Filling amount values are the means of the replicates. Means with different letters are significantly different.

**Table 2 foods-11-03515-t002:** The phytochemical composition of “Lollo Bionda” lettuce and cultivated rocket at harvest (d0).

	Chl. *a* (mg g^−1^ FW)	Chl. *b* (mg g^−1^ FW)	Car. (mg g^−1^ FW)	TP (mg Gallic Acid g^−1^ FW)	AA (mg g^−1^ FW)	PAL (µmol Cinnamic Acid h^−1^ g^−1^ FW)	PO_4_^3−^ (mg g^−1^ FW)	CaCO_3_ (mg g^−1^ FW)
Species								
LB	0.20 b	0.06 b	0.09 b	0.26 b	0.23 b	0.63 b	0.14 a	0.55 b
RO	0.53 a	0.15 a	0.21 a	0.70 a	0.38 a	2.36 a	0.07 b	2.16 a
Mean	0.37	0.11	0.15	0.48	0.31	1.49	0.11	1.35
Significance	*0.002*	*0.001*	*0.002*	*0.001*	*0.022*	*0.016*	*0.020*	*<0.001*

LB = ‘Lollo Bionda’; RO = cultivated rocket; Chl. *a* = chlorophyll *a* content; Chl. *b* = chlorophyll *b* content; Car. = carotenoid content; TP = total phenolic content; AA = ascorbic acid content; PAL = phenylalanine ammonia lyase activity; PO_4_^3−^ = phosphate content; CaCO_3_ = calcium carbonate content. ^z^ Mean separation (in columns) by the F-test at *p* ≤ 0.05. Means with different letters are significantly different.

**Table 3 foods-11-03515-t003:** Antioxidant properties of the fresh-cut baby leaf lettuce and rocket during shelf-life.

	AC (μmol Fe^2+^ g^−1^ FW)		TP (mg Gallic Acid g^−1^ FW)		AA (mg g^−1^ FW)		DHAA (mg g^−1^ FW)	
	d1	d9	d1	d9	d1	d9	d1	d9
Mix								
100 LB	2.49 b ^z^	2.07	0.22 b	0.16 c	0.219 c	0.115 c	0.030 b	0.068 b
75 LB	2.68 b	2.10	0.25 b	0.26 b	0.307 b	0.164 b	0.035 ab	0.064 b
50 LB	3.44 a	2.39	0.37 a	0.37 a	0.330 a	0.271 a	0.040 a	0.083 a
Filling amount								
F250	2.99	2.17	0.30	0.22 b	0.296 a	0.170 b	0.036	0.071
F125	2.75	2.20	0.26	0.31 a	0.275 b	0.197 a	0.034	0.072
Mix × Filling amount								
100 LB × F250	2.84	2.09	0.25	0.15	0.227	0.111	0.035	0.072
100 LB × F125	2.13	2.05	0.18	0.17	0.211	0.120	0.025	0.065
75 LB × F250	2.62	2.27	0.28	0.25	0.324	0.146	0.035	0.062
75 LB × F125	2.73	1.93	0.23	0.27	0.290	0.183	0.035	0.066
50 LB × F250	3.50	2.15	0.36	0.26	0.337	0.254	0.037	0.079
50 LB × F125	3.39	2.63	0.37	0.48	0.323	0.288	0.043	0.087
Mean	2.87	2.19	0.28	0.26	0.285	0.183	0.035	0.072
SE	0.19	0.13	0.02	0.03	0.006	0.011	0.003	0.005
Significance								
Mix	*<0.001*	0.058	*<0.001*	*<0.001*	*<0.001*	*<0.001*	*0.021*	*0.002*
Filling amount	0.130	0.764	0.085	*<0.001*	*<0.001*	*0.005*	0.563	0.734
Mix × Filling amount	0.092	*0.017*	0.232	*0.001*	0.258	0.372	0.082	0.291

d1 = 1 day of storage; d9 = 9 days of storage; LB = Lollo Bionda; F = grams of the filling amount in the fresh-cut salad bag; AC = antioxidant capacity; TP = total phenolic content; AA = ascorbic acid content; DHAA = dehydroascorbic acid content; SE = standard error. ^z^ Mean separation (in columns) by Tukey’s multiple range test at *p* ≤ 0.05 for the Mix, and by the F-test at *p* ≤ 0.05 for the Filling amount. The Mix × Filling amount values are the means of the replicates. Means with different letters are significantly different.

**Table 4 foods-11-03515-t004:** Tissue browning parameters of the fresh-cut baby leaf lettuce and rocket during shelf-life.

	BP (Abs_340_)		So-Q (Abs_437_)		PAL (µmol Cinnamic Acid h^−1^ g^−1^ FW)		PPO (Units g^−1^ FW)	
	d1	d9	d1	d9	d1	d9	d1	d9
Mix								
100 LB	0.15 c	0.17 c	0.10 c	0.10 b	0.83 b	0.97 c	18.41 ab ^z^	25.30 b
75 LB	0.35 b	0.32 b	0.12 b	0.14 ab	1.12 b	1.45 b	17.80 b	25.23 b
50 LB	0.68 a	0.69 a	0.17 a	0.21 a	2.19 a	2.28 a	20.38 a	27.80 a
Filling amount								
F250	0.41	0.33 b	0.12	0.13	1.46	1.31 b	19.14	28.54 a
F125	0.37	0.45 a	0.13	0.18	1.30	1.83 a	18.59	23.69 b
Mix × Filling amount								
100 LB × F250	0.15	0.14	0.09	0.08	0.79	0.97	19.39	29.55
100 LB × F125	0.14	0.19	0.10	0.13	0.88	0.97	17.42	21.06
75 LB × F250	0.35	0.24	0.12	0.09	1.20	1.11	17.12	26.82
75 LB × F125	0.34	0.40	0.13	0.20	1.04	1.79	18.48	23.64
50 LB × F250	0.72	0.61	0.16	0.21	2.40	1.85	20.91	29.24
50 LB × F125	0.64	0.77	0.17	0.20	1.99	2.71	19.85	26.36
Mean	0.39	0.39	0.13	0.15	1.38	1.57	18.86	26.11
SE	0.03	0.04	0.01	0.03	0.12	0.17	1.02	0.79
Significance								
Mix	*<0.001*	*<0.001*	*<0.001*	*0.005*	*<0.001*	*<0.001*	*0.045*	*0.004*
Filling amount	0.142	*<0.001*	0.141	0.053	0.133	*0.001*	0.512	*<0.001*
Mix × Filling amount	0.294	0.252	0.930	0.176	0.158	*0.041*	0.260	*0.002*

d1 = 1 day of storage; d9 = 9 days of storage; LB = ‘Lollo Bionda’; F = grams of the filling amount in the fresh-cut salad bag; BP = browning potential; So-Q = soluble o-quinone content; PAL = phenylalanine ammonia lyase activity; PPO = polyphenol oxidase activity; SE = standard error. ^z^ Mean separation (in columns) by Tukey’s multiple range test at *p* ≤ 0.05 for the Mix, and by the F-test at *p* ≤ 0.05 for the Filling amount. The Mix × Filling amount values are the means of the replicates. Means with different letters are significantly different.

**Table 5 foods-11-03515-t005:** Ion and salt contents of the tissue, and microbial contamination in the fresh-cut baby leaf lettuce and rocket during shelf-life.

	NO_3_^−^ (mg g^−1^ FW)		PO_4_^3−^ (mg g^−1^ FW)		CaCO_3_ (mg g^−1^ FW)		TB (cfu g^−1^ FW)	Y + M (cfu g^−1^ FW)
	d1	d9	d1	d9	d1	d9	d9	d9
Mix								
100 LB	0.49 ^z^	0.28 b	0.14 a	0.15 a	0.47 c	0.35 c	5.91 × 10^3^ b	4.63 × 10^1^ b
75 LB	0.44	0.28 b	0.13 a	0.13 b	0.93 b	0.71 b	7.74 × 10^4^ a	8.00 × 10^1^ a
50 LB	0.43	0.36 a	0.10 b	0.10 c	1.22 a	1.22 a	4.62 × 10^4^ ab	1.01 × 10^2^ a
Filling amount								
F250	0.45	0.30	0.12	0.13	0.86	0.69 b	2.65 × 10^4^	8.43 × 10^1^
F125	0.46	0.32	0.13	0.12	0.89	0.83 a	5.98 × 10^4^	6.69 × 10^1^
Mix × Filling amount								
100 LB × F250	0.47	0.30	0.13	0.15	0.43	0.30	6.82 × 10^3^	5.53 × 10^1^
100 LB × F125	0.51	0.26	0.15	0.15	0.51	0.40	5.00 × 10^3^	3.73 × 10^1^
75 LB × F250	0.46	0.27	0.13	0.14	0.89	0.57	3.63 × 10^4^	7.93 × 10^1^
75 LB × F125	0.41	0.29	0.14	0.11	0.96	0.84	1.18 × 10^5^	8.07 × 10^1^
50 LB × F250	0.41	0.33	0.10	0.09	1.25	1.20	3.64 × 10^4^	1.18 × 10^2^
50 LB × F125	0.45	0.39	0.11	0.10	1.19	1.23	5.59 × 10^4^	8.27 × 10^1^
Mean	0.45	0.31	0.13	0.12	0.87	0.76	4.31 × 10^4^	7.56 × 10^1^
SE	0.04	0.03	0.01	0.01	0.07	0.06	2.03 × 10^4^	1.20 × 10^1^
Significance								
Mix	0.252	*0.005*	*0.001*	*<0.001*	*<0.001*	*<0.001*	*0.006*	*<0.001*
Filling amount	0.727	0.457	0.068	0.066	0.649	*0.011*	0.055	0.085
Mix × Filling amount	0.437	0.125	0.479	*0.049*	0.578	0.132	0.119	0.318

d1 = 1 day of storage; d9 = 9 days of storage; LB = ‘Lollo Bionda’; F = grams of the filling amount in the fresh-cut salad bag; NO_3_^−^ = nitrate content; PO_4_^3−^ = phosphate content; CaCO_3_ = calcium carbonate content; TB = total bacterial count; Y + M = yeast + mould count; FW = fresh weight; SE = standard error. ^z^ Mean separation (in columns) by Tukey’s multiple range test at *p* ≤ 0.05 for the Mix, and by the F-test at *p* ≤ 0.05 for the Filling amount. The Mix × Filling amount values are the means of the replicates. Means with different letters are significantly different.

## Data Availability

The data presented in this study are available on request from the corresponding author.

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
