# Peer review of "The Mixing Ratio and Filling-Amount Affect the Tissue Browning and Antioxidant Properties of Fresh-Cut Baby Leaf Lettuce (Lactuca sativa L.) and Rocket (Eruca sativa Mill.) Grown in Floating Growing Systems"

_foods, 2022, doi:10.3390/foods11213515_

Round 1

Reviewer 1 Report

The paper "The mixing ratio and filling-amount affect the tissue browning and antioxidant properties of fresh-cut baby leaf lettuce (Lactuca sativa L.) and rocket (Eruca sativa Mill.) grown in floating growing systems" is interesting and well organized.

Some points should be clarified:
-Please, explain better the synergistic effect of the different leaf types. Is it only all about the respiratory issues? Add other reference studies.
-It should be relevant to add quality evaluation of the raw material, of the individual and mix species (representative sample), after harvest.
- Did the authors perform any hygienic procedure of the raw material after harvest?
Please add information on the quality parameters evaluated. The information presented is too short. The authors should add basic but relevant information, for example:  the amount of sample used in each procedure; ratios of mass /volume, unit in enzymatic determination.
-What is the meaning of the mean and SE for example in Table 2 and 3?
-Temperature effect is a very relevant parameter to preserve the quality of these types of products. Why did the authors choose 4ºC to store the product and only considered 9d of storage?

-This study presents a major issue. It doesn´t evaluate each vegetable per se.

Author Response

REV 1: Comments and Suggestions for Authors

The paper "The mixing ratio and filling-amount affect the tissue browning and antioxidant properties of fresh-cut baby leaf lettuce (Lactuca sativa L.) and rocket (Eruca sativa Mill.) grown in floating growing systems" is interesting and well organized.

Some points should be clarified:

-Please, explain better the synergistic effect of the different leaf types. Is it only all about the respiratory issues? Add other reference studies.

A.A.: Apart from respiratory issues, the favourable synergistic effect of mixing ratio and filling amount (50 LB×125F) is mainly associated with the high initial product quality (50 LB samples showed high RO ratio, and RO had a high quality, at harvest) and with less injury of leaves (125F suffers less crush injury due to larger headspace). These aspects have been added in the text. Furthermore, we also hypothesized that the addition in the mix of rocket, belonging to the Brassicacea family, could increase the shelf-life of the product (for example due to exudates, bioactive molecules), an aspect that would be interesting to deepen in future experiments.

Rocket leaves were already evaluated in our previous works:

Nicola, S., Fontana, E., Tibaldi, G. & Zhan, L. Qualitative and physiological response of minimally processed rocket (Eruca sativa Mill.) to package filling amount and shelf-life temperature. Acta Hort, 2010; 877, 611-618.

Nicola, S., G. Pignata, and G. Tibaldi. The floating growing system can assure a low microbial contamination of baby leaf vegetables at harvest. Acta Hort, 2018; 1209, 57-64.

-It should be relevant to add quality evaluation of the raw material, of the individual and mix species (representative sample), after harvest.

A.A.: The quality evaluation of raw material of each individual species was performed at harvest, as showed in Table 2, through the study of chlorophylls and carotenoids, total phenols, AA, nitrate, and other important BLV quality parameters (only the parameters with significant differences were listed, and those without significant differences were also described in the Results section).  For the raw material of mix species, the quality evaluation was done from 1 day of shelf-life (d1), but not at harvest (d0) because, according to our experience, the leaf quality change in an irrelevant way after 1 day of storage at 4 ºC in the dark.

We thank the reviewer for the constructive comment that we will take into account for future experiments so as to be more accurate in showing the quality dynamic changes of mixed samples after harvest.

- Did the authors perform any hygienic procedure of the raw material after harvest?

A.A.: As reported in the text, at harvest we used sanitised tools. No washing has been carried out on the leaves because, as also reported in the manuscript, the TB count and Y+M count of the samples were below the legal limits. In addition, as indicated in the M&M section, our product has been grown using soilless cultivation systems, that allow obtaining a cleaner and hygienically safer product (Nicola et al., 2018).

Please add information on the quality parameters evaluated. The information presented is too short. The authors should add basic but relevant information, for example:  the amount of sample used in each procedure; ratios of mass /volume, unit in enzymatic determination.

A.A.:  The requested information has been added in the text, in the M&M section.

-What is the meaning of the mean and SE for example in Table 2 and 3?

A.A.: The mean is the average value, and SE is the abbreviation of standard error. We added this information in the Tables.

-Temperature effect is a very relevant parameter to preserve the quality of these types of products. Why did the authors choose 4ºC to store the product and only considered 9d of storage?

A.A.: We totally agree with the Reviewer 1 on the importance of temperature in preserving the quality of BLVs. In this regard, we have chosen the optimal temperature for their preservation (4°C), following the good manufacturing practices, the US FDA, and as also reported in several published experiments (Spinardi and Ferrante, 2012; Cavaiuolo et al., 2015; Saini et al., 2016; Tudela et al., 2017; Bulgari et al., 2017; Nicola et al., 2022). The temperature is known to be the single most important factor affecting the quality of fresh-cut vegetables in post-harvest, in order to maintain a good visual appearance, quality, and safety (4℃ limit various microbial activities). For all these reasons the research data of the present work can be used in practice.

All the qualitative parameters were measured after 1 day of shelf-life (d1) and at the end of the shelf-life (d9). Currently, fresh-cut vegetables shelf-life is ca. 7 days in Italy and in many EU countries. In our research, we found that the samples still showed a good quality after 9 days. So, we considered the time point of 9 days to evaluate a long shelf-life of the product. Being able to extend the limit of marketability could have interesting implications, both economic and in terms of food waste reduction. The results have shown that the shelf-life was extended for 2 days compared to the standard 7 days, which proved that the innovative synergistic strategy we adopted in this study is effective and feasible, and has a good potential prospect in improving the economic value of fresh-cut products.

-This study presents a major issue. It doesn´t evaluate each vegetable per se.

A.A: This research is based on market demand. At present, in most fresh-cut mixed salad products, rocket leaves are often mixed as a minor ingredient to provide itself special spicy flavor. Increasing rocket leaves ratio (>50%) will increase the strong spicy taste of mixed salad, and thus it can reduce the mixed salad palatability. In addition, increasing rocket leaves ratio in mixed salad will increase the product cost, as rocket leaves have higher price compared to other salads, such as lettuce, cabbage, etc.

More specifically, lettuce (Lactuca sativa L.) is one of the major vegetables consumed as salad in the human diet and economically important vegetables; it represents a significant proportion of this ready-to-eat salad market (Spinardi and Ferrante, 2012; Kim et al., 2016; Rocha et al., 2020; Damerum et al., 2020; Martínez-Ispizua et al., 2022). In particular, baby lettuce leaves with their petiole, at an optimum size of 8–12 cm long, have become very popular in recent years as minimally processed or ready-to-eat fresh-cut salad vegetables. They represent an innovative produce, corresponding to the lifestyle of modern consumers (Gil et al., 2020; Carillo et al., 2020). Most salads, mainly prepared ones, continue to be mainly composed of lettuce.

Rocket is well known for its spicy flavor and has a continuously increasing demand. Especially baby rocket leaves are a major economic product, as they are a basic ingredient of ready-to-eat salads in combination with other leafy vegetables (Koukounaras et al., 2020).

We already evaluated rocket leaves per se in our previous work (Nicola, S., Fontana, E., Tibaldi, G. & Zhan, L. Qualitative and physiological response of minimally processed rocket (Eruca sativa Mill.) to package filling amount and shelf-life temperature. Acta Hort, 2010; 877, 611-618). This research allowed us to define the quality and shelf-life of the product, to plan future tests on the study of mixes.

Reviewer 2 Report

Peer-review on the article “The mixing ratio and filling-amount affect the tissue browning and antioxidant properties of fresh-cut baby leaf lettuce (Lactuca sativa L.) and rocket (Eruca sativa Mill.) grown in floating growing systems. written by Lijuan Zhan, Roberta Bulgari, Giuseppe Pignata, Manuela Casale and Silvana Nicola to be published in Foods.

This paper discusses the optimal postharvest strategies for the mixture ration and packing filling amount for the fresh-cut chain of fresh-cut baby leaf lettuce (Lactuca sativa L.) and rocket (Eruca sativa Mill.) salad mixes. A mixing ratio of 50% lettuce and 50% rocket has been found to favor the preservation of chlorophyll and the internal nutrients at the end of storage and no visible browning symptoms were detected. Overall, this paper also suggests efficient cultural practices to obtain high quality baby leaf vegetables before harvesting, besides the optimal postharvest strategies for the mixture ration and packing filling amount for the fresh-cut chain.

Minor comments:

It is mentioned in 2. Materials and Methods that the green leaf lettuce (Lactuca sativa L. var. crispa cv. Lollo Bionda is hereafter LB and the cultivated rocket is hereafter RO. I think this needs to be mentioned earlier, probably after the first sentence explaining that the term BLV is used to define baby leaf vegetables. it is mentioned in the Table 5., that LB = percentage of “’Lollo Bionda” in the mix with rocket.

Modified atmosphere packaging (MAP) and passive modified atmosphere packaging (passive MAP) is explained in 1. Introduction. But it seems to be reasonable to mention in the abstract that the packaging and storage conditions in the experiment involved one of these technologies. The samples were packaged in thermos-sealed bags previously prepared using polypropylene film with the same permeance to oxygen, carbon dioxide and to water vapor. The packaged samples were stored at 4 °C for 9 days, in refrigerated chambers without light.

The chlorophyll a (Chl. a), chlorophyll b (Chl. b) and carotenoid (Car.) contents were determined according to the protocol suggested in literatures from 2009 and 1983. Soil Plant Analysis Development (SPAD) value is related to the chlorophyll concentration in the leaves. Since the chlorophyll content has a high correlation with the concentration of carotenoids in lettuce, it can be used to assess the amount of carotenoids in lettuce leaves through the SPAD index (Cassetari et al., 2015; Maciel et al., 2019; Peixoto et al., 2021). Baby leaf vegetables are the rich source of vitamins, minerals and antioxidant phytochemicals, including carotenoids, tocopherols and folates (Lopez et al. 2014). Carotenoids and tocopherols are the primary fat-soluble vitamins that have epidemiological evidence of benefiting human health (Ju et al. 2010). The baby leaf vegetable crops with higher SPAD values tended to have thicker leaves according to a Japanese study (Takahama et al., 2019).

Further types of baby leaf vegetables used in salad mixes needs to be involved in these kinds of studies. Of all the studied varieties, the dark red, almost purple landrace L11 (Genbank code) lettuce (Lactuca sativa L.) variety was significant in all the studied stages and in almost all the analyzed parameters in a recent study in Valencia, Spain. Ascorbic acid and phenolic contents were 42% and 79% higher, respectively, in the early stages than in adult lettuces, and red-leaf varieties (CL4 and L11) stood out (Martínez-Ispizua et al., 2022).

Literature

Cassetari, L. S., Gomes, M. S., Santos, D. C., Santiago, W. D., Andrade, J., Guimaraes, A. C., Souza, J. A., Caedoso, M. G.; Maluf, W. R.; Gomes, L. A. (2015): β-carotene and chlorophyll levels in cultivars and breeding lines of lettuce. Acta Horticulturae, 1083, 469-473.

Ju, J., Picinich, S. C., Yang, Z., Zhao, Y., Suh, N., Kong, A-N., Yang, C. S. (2010): Cancer preventive activities of tocopherols and tocotrienols. Carcinogenesis, 31, 533-542.

Lopez, A., Javier, G-A., Fenoll, J., Hellin, P., Flores, P. (2014): Chemical composition and antioxidant capacity of lettuce: comparative study of regular-sized (Romaine) and baby sized (Little Gem and Mini Romaine) types. Journal of Food Composition and Analysis, 33, 39–48.

Maciel, G. M., Gallis, R. B. A., Barbosa, R. L., Pereira, L. M., Siquieroli, A. C. S., Peixoto, J. V. M. (2019): Image phenotyping of inbred red lettuce lines with genetic diversity regarding carotenoid levels. International Journal of Applied Earth Observation and Geoinformation, 81, 154-160.

Martínez-Ispizua, E., Calatayud, Á., Marsal, J. I., Cannata, C., Basile, F., Abdelkhalik, A., Soler, S., Valcárcel, J. V., Martínez-Cuenca, M. -R. (2022): The Nutritional Quality Potential of Microgreens, Baby Leaves, and Adult Lettuce: An Underexploited Nutraceutical Source. Foods, 11, 423.

Peixoto, J. V. M., Maciel, G. M., Finzi, R. R.; Pereira, L. M.; Siquieroli, A. C. S., Silva, M. F., Clemente, A. A. (2021): Genetic parameters and selection indexes for biofortified red leaf lettuce. Pesquisa Agropecuária Brasileira, 56, e02431.

Saini, R. K., Shang, X. M., Ko, E. Y., Choi, J. H., Keum, Y. S. (2015): Stability of carotenoids and tocopherols in ready-to-eat baby-leaf lettuce and salad rocket during low temperature storage. International Journal of Food Science and Nutrition, 67, 5, 489-495.

Takahama, M., Kawagishi, K., Sugawara, A., Araki, K., Munekata, S., Nicola S., Araki, H. (2019): Classification and Screening of Baby-leaf Vegetables on the Basis of Their Yield, External Appearance and Internal Quality. The Horticulture Journal 88, 3, 387–400.

Author Response

REV 2: Comments and Suggestions for Authors

Peer-review on the article “The mixing ratio and filling-amount affect the tissue browning and antioxidant properties of fresh-cut baby leaf lettuce (Lactuca sativa L.) and rocket (Eruca sativa Mill.) grown in floating growing systems.” written by Lijuan Zhan, Roberta Bulgari, Giuseppe Pignata, Manuela Casale and Silvana Nicola to be published in Foods.

This paper discusses the optimal postharvest strategies for the mixture ration and packing filling amount for the fresh-cut chain of fresh-cut baby leaf lettuce (Lactuca sativa L.) and rocket (Eruca sativa Mill.) salad mixes. A mixing ratio of 50% lettuce and 50% rocket has been found to favor the preservation of chlorophyll and the internal nutrients at the end of storage and no visible browning symptoms were detected. Overall, this paper also suggests efficient cultural practices to obtain high quality baby leaf vegetables before harvesting, besides the optimal postharvest strategies for the mixture ration and packing filling amount for the fresh-cut chain.

Minor comments:

It is mentioned in 2. Materials and Methods that the green leaf lettuce (Lactuca sativa L. var. crispa cv. Lollo Bionda is hereafter LB and the cultivated rocket is hereafter RO. I think this needs to be mentioned earlier, probably after the first sentence explaining that the term BLV is used to define baby leaf vegetables. it is mentioned in the Table 5., that LB = percentage of “’Lollo Bionda” in the mix with rocket.

A.A.: Thank you for your suggestions. The abbreviations used (LB and RO) have been revised in the text and tables’ notes.

Modified atmosphere packaging (MAP) and passive modified atmosphere packaging (passive MAP) is explained in 1. Introduction. But it seems to be reasonable to mention in the abstract that the packaging and storage conditions in the experiment involved one of these technologies. The samples were packaged in thermos-sealed bags previously prepared using polypropylene film with the same permeance to oxygen, carbon dioxide and to water vapor. The packaged samples were stored at 4 °C for 9 days, in refrigerated chambers without light.

A.A.: This is a useful suggestion. Unfortunately, the abstract has a defined length (200 words maximum), and therefore we had to choose the most important information to report. We made some changes to the abstract, trying to include the information suggested by Reviewer 2. Thanks for the suggestion.

The chlorophyll a (Chl. a), chlorophyll b (Chl. b) and carotenoid (Car.) contents were determined according to the protocol suggested in literatures from 2009 and 1983. Soil Plant Analysis Development (SPAD) value is related to the chlorophyll concentration in the leaves. Since the chlorophyll content has a high correlation with the concentration of carotenoids in lettuce, it can be used to assess the amount of carotenoids in lettuce leaves through the SPAD index (Cassetari et al., 2015; Maciel et al., 2019; Peixoto et al., 2021). Baby leaf vegetables are the rich source of vitamins, minerals and antioxidant phytochemicals, including carotenoids, tocopherols and folates (Lopez et al. 2014). Carotenoids and tocopherols are the primary fat-soluble vitamins that have epidemiological evidence of benefiting human health (Ju et al. 2010). The baby leaf vegetable crops with higher SPAD values tended to have thicker leaves according to a Japanese study (Takahama et al., 2019).

A.A.: We thank the Reviewer 2 for suggestions and insights on the topic. We did not use in vivo measurements (SPAD) in our experiment because we wanted to study the pigment concentrations (mg g-1 FW). We added a sentence in the M&M section. SPAD is a measurement of the leaf’s green coloration (greenness). We certainly recognize the usefulness of this tool, and we will keep it in mind in the future, to correlate the data obtained through destructive and non-destructive measurements. Since this research is not only based on the cultivation of vegetables, but also analyzes the leaves obtained from a nutritional point of view, nutrients types and accurate contents are necessary in order to provide suggestions for healthy diet, which is different from comparing the relative value of chlorophyll with SPAD in plant cultivation.

Further types of baby leaf vegetables used in salad mixes needs to be involved in these kinds of studies. Of all the studied varieties, the dark red, almost purple landrace L11 (Genbank code) lettuce (Lactuca sativa L.) variety was significant in all the studied stages and in almost all the analyzed parameters in a recent study in Valencia, Spain. Ascorbic acid and phenolic contents were 42% and 79% higher, respectively, in the early stages than in adult lettuces, and red-leaf varieties (CL4 and L11) stood out (Martínez-Ispizua et al., 2022).

A.A.: Thank you for the suggestion. Certainly the study of other types of salad mix is interesting, and we have evaluated other types of mix with leafy vegetables in previous studies, as reported, for example, in the Introduction section: “One of our previous studies, in fact, showed that the mixing ratio of fresh-cut baby lettuce with 50 LB (50% green lettuce, 50% red lettuce) and 25 LB (25% green lettuce, 75% red lettuce) had…” Pignata et al., 2020. We added a sentence in the Conclusions section to highlight this suggestion.

Literature

A.A.: thanks for all the suggested papers; we added the most relevant ones within the text.
